# Catalyst: Fast and flexible modeling of reaction networks

**Torkel E. Loman** [1,2]☙ *, **Yingbo Ma**[3], **Vasily Ilin**[4], **Shashi Gowda**[5], **Niklas Korsbo**[6], **Nikhil Yewale**[7], **Chris Rackauckas**[2,3,6]☙ *, **Samuel A. Isaacson**[8]☙ *

**1** Sainsbury Laboratory, University of Cambridge, Cambridge, United Kingdom, **2** Computer Science and AI Laboratory (CSAIL), Massachusetts Institute of Technology, Cambridge, Massachusetts, United States of America, **3** JuliaHub, Cambridge, Massachusetts, United States of America, **4** Department of Mathematics, University of Washington, Seattle, Washington, United States of America, **5** Department of Mathematics, Massachusetts Institute of Technology, Cambridge, Massachusetts, United States of America, **6** Pumas-AI, Baltimore, Maryland, United States of America, **7** Department of Applied Mechanics, Indian Institute of Technology Madras, Chennai, India, **8** Department of Mathematics and Statistics, Boston University, Boston, Massachusetts, United States of America

☙ These authors contributed equally to this work.
* torkell@mit.edu (TEL); crackauc@mit.edu (CR); isaacson@math.bu.edu (SAI)

**Data Availability Statement:** Scripts for generating all figures presented here, as well as for carrying out the benchmarks, can be found at https://github.com/SciML/Catalyst_PLOS_COMPBIO_2023, with an archived version available at https://doi.org/10.5281/zenodo.8364792. Catalyst is available for free

## Abstract

We introduce Catalyst.jl, a flexible and feature-filled Julia library for modeling and high-performance simulation of chemical reaction networks (CRNs). Catalyst supports simulating stochastic chemical kinetics (jump process), chemical Langevin equation (stochastic differential equation), and reaction rate equation (ordinary differential equation) representations for CRNs. Through comprehensive benchmarks, we demonstrate that Catalyst simulation runtimes are often one to two orders of magnitude faster than other popular tools. More broadly, Catalyst acts as both a domain-specific language and an intermediate representation for symbolically encoding CRN models as Julia-native objects. This enables a pipeline of symbolically specifying, analyzing, and modifying CRNs; converting Catalyst models to symbolic representations of concrete mathematical models; and generating compiled code for numerical solvers. Leveraging ModelingToolkit.jl and Symbolics.jl, Catalyst models can be analyzed, simplified, and compiled into optimized representations for use in numerical solvers. Finally, we demonstrate Catalyst's broad extensibility and composability by highlighting how it can compose with a variety of Julia libraries, and how existing open-source biological modeling projects have extended its intermediate representation.

## Author summary

Chemical reaction networks (CRNs) are a type of model commonly used in biology and chemistry. Their applications include the investigation of cellular system functions (systems biology), designing drugs (pharmacology), and forecasting epidemic progression (epidemiology). In this article, we present the Catalyst.jl software for the modelling, simulation, and analysis of CRNs across several physical scales. Catalyst simulations of CRN models are often one to two orders of magnitude faster than other popular CRN modeling

under the permissive MIT License. The source code can be found at https://github.com/SciML/Catalyst.jl. It is also a registered package within the Julia ecosystem and can be installed from within a Julia environment using the commands using Pkg; Pkg.add("Catalyst"). Full documentation, including tutorials and an API, can be found at https://catalyst.sciml.ai/stable/. Issues and help requests can be raised either at the Catalyst GitHub page, on the Julia discourse forum (https://discourse.julialang.org/), or at the SciML organization's Julia language Slack channels (#diffeq-bridged and #sciml-bridged). The library is open to pull requests from anyone who wishes to contribute to its development. Users are encouraged to engage in the project.

**Funding:** TL's contribution to this project has received funding from the European Union's Horizon 2020 research and innovation programme under the Marie Sklodowska-Curie grant agreement No.721456. TL received salary support from the aforementioned funding source. SAI's and CR's work on this project has been made possible in part by the following two grants to the SciML organization. This research was funded in whole, or in part, by the Wellcome Trust [223770/Z/21/Z]. For the purpose of open access, the author has applied a CC BY public copyright licence to any Author Accepted Manuscript version arising from this submission. This publication and software have been made possible in part by CZI grant DAF2021-237457 and grant DOI https://doi.org/10.37921/149019qvrhgz from the Chan Zuckerberg Initiative DAF, an advised fund of Silicon Valley Community Foundation (funder DOI 10.13039/100014989). SAI was also partially supported by National Science Foundation DMS-1902854. VI was partially supported by a 2021 Google Summer of Code Fellowship and the Boston University UROP program. SAI and VI each received salary support from each of their respective aforementioned funding sources. CR's contribution to this material is based upon work supported by the National Science Foundation under grant no. OAC-1835443, grant no. SII-2029670, grant no. ECCS-2029670, grant no. OAC-2103804, and grant no. PHY-2021825. We also gratefully acknowledge the U.S. Agency for International Development through Penn State for grant no. S002283-USAID. The information, data, or work presented herein was funded in part by the Advanced Research Projects Agency-Energy (ARPA-E), U.S. Department of Energy, under Award Number DE-AR0001211 and DE-AR0001222. This material is based upon work supported by the Defense Advanced Research Projects Agency (DARPA) under Agreement No HR00112290091. We also

tools. Such speed increases in turn aid in facilitating a variety of CRN analyses, for example simulating a model across a large number of conditions to check which ones best fit real-world observations. Catalyst also includes a domain-specific modeling language, which allows users to easily input their CRN models using standard chemical reaction syntax, thereby simplifying model creation. Finally, Catalyst is built on top of a widely used symbolic computer algebra system and mathematical modeling framework. This allows significant flexibility in the types of components that can be used within Catalyst CRN models, has enabled the development of tools to analyze and transform Catalyst models, and has helped Catalyst to become a broadly extensible package that can be composed with a variety of independent software libraries.

# 1 Introduction

Chemical reaction network (CRN) models are used across a variety of fields, including the biological sciences, epidemiology, physical chemistry, combustion modeling, and pharmacology [1–7]. At their core, they combine a set of species (defining a system's state) with a set of reaction events (rates for reactions occurring combined with rules for altering the system's state when a reaction occurs). One advantage of formulating a biological model as a CRN is that these can be simulated according to several well-defined mathematical representations, representing different physical scales at which reaction processes can be studied. For example, the *reaction rate equation* (RRE) is a macroscopic system of ordinary differential equations (ODEs), providing a deterministic model of chemical reaction processes. Similarly, the *chemical Langevin equation* (CLE) is a system of stochastic differential equations (SDEs), providing a more microscopic model that can capture certain types of fluctuations in reaction processes [8]. Finally, stochastic chemical kinetics, typically simulated with the *Gillespie algorithm* (as well as modifications to, and improvements of, it), provides an even more microscopic model, that captures both stochasticity and discreteness of populations in chemical reaction processes [9, 10]. That a CRN can be unambiguously represented using these models forms the basis of several CRN modeling tools [11–26]. Here we present a new modeling tool for CRNs, Catalyst.jl, which we believe offers a unique set of advantages for both inexperienced and experienced modelers.

Catalyst's defining trait, which sets it apart from other popular CRN modeling packages, is that it represents models in an entirely symbolic manner, accessible via standard Julia language programs. This permits algebraic manipulation and simplification of the models, either by the user, or by other tools. Once a CRN has been defined, it is stored in a symbolic *intermediate representation* (IR). This IR is the target of methods that provide functionality to Catalyst, including numerical solvers for both continuous ODEs and SDEs, as well as discrete Gillespie-style stochastic simulation algorithms (SSAs). As Catalyst's symbolic representations can be converted to compiled Julia functions, it can be easily used with a variety of Julia libraries. These include packages for parameter fitting, sensitivity analysis, steady state finding, and bifurcation analysis. To simplify model implementation, Catalyst provides a *domain-specific language* (DSL) that allows users to declare CRN models using classic chemical reaction notation. Finally, Catalyst also provides a comprehensive API to enable programmatic manipulation and combination of models, combined with functionality for analyzing and simplifying CRNs (such as detection of conservation laws and elimination of conserved species).

Catalyst is implemented in Julia, a relatively recent (version 1.0 released in August 2018) open-source programming language for scientific computing. Its combination of high

gratefully acknowledge the U.S. Agency for International Development through Penn State for grant no. S002283-USAID. The views and opinions of authors expressed herein do not necessarily state or reflect those of the United States Government or any agency thereof. This material was supported by The Research Council of Norway and Equinor ASA through Research Council project "308817 - Digital wells for optimal production and drainage". Research was sponsored by the United States Air Force Research Laboratory and the United States Air Force Artificial Intelligence Accelerator and was accomplished under Cooperative Agreement Number FA8750-19-2-1000. The views and conclusions contained in this document are those of the authors and should not be interpreted as representing the official policies, either expressed or implied, of the United States Air Force or the U.S. Government. The U.S. Government is authorized to reproduce and distribute reprints for Government purposes notwithstanding any copyright notation herein. The funders had no role in study design, data collection and analysis, decision to publish, or preparation of the manuscript.

**Competing interests:** I have read the journal's policy and the authors of this manuscript have the following competing interests: YM and CR are employed by JuliaHub, a cloud computing company specialising in Julia applications. NK and CR are employed by Pumas-AI, a company developing Julia-based software platforms for the pharmaceutical industry.

performance and user-friendliness makes it highly promising [27, 28]. Julia has grown quickly, with a highly developed ecosystem of packages for scientific simulation. This includes the many packages provide by the Scientific Machine Learning (SciML) organization, of which Catalyst is a part. SciML, through its ModelingToolkit.jl package, provides the IR on which Catalyst is based [29]. This IR is used across the organization's projects, providing a target structure both for model-generation tools (such as Catalyst), and tools that provide additional functionality. ModelingToolkit symbolic models leverage the Symbolics.jl [30] *computer algebraic system* (CAS), enabling them to be represented in a symbolic manner. Simulations of ModelingToolkit-based models are typically carried out using DifferentialEquations.jl, perhaps the largest software package of state-of-the-art, high-performance numerical solvers for ODEs, SDEs, and jump processes [31].

Several existing modeling packages provide overlapping functionality with Catalyst. COPASI is a well known and popular software that enables both deterministic and stochastic CRN modeling, as well as many auxiliary features (such as parameter fitting and sensitivity analysis) [12]. BioNetGen is another such software suite, currently available as a Visual Studio Code extension, that is built around the popular BioNetGen language for easily specifying complex reaction network models [21]. It provides options for model creation, network simulation, and network free-modeling. Another popular tool, VCell, provides extensive functionality, via an intuitive graphical interface [11]. Finally, Tellurium ties together a range of tools to be used in a Python environment, allowing CRN models to be created using the Antimony DSL, and simulated using the libRoadRunner numeric solver suite [15, 23, 24, 32]. Other modeling softwares include GINsim, CellNOpt, GillesPy2, and Matlab's SimBiology toolbox [13, 16, 33].

Several of these packages are primarily designed around a GUI-based workflow (BioNetGen, COPASI, and VCell). In contrast, Catalyst is DSL and API-based, with simulation and analysis of models carried out via Julia scripts. A typical Catalyst workflow therefore requires users to write Julia language scripts instead of using a GUI-based interface, but also enables users to easily integrate Catalyst models with a large variety of other Julia libraries. Catalyst also has immediate access to a more extensive set of numerical solvers for ODEs, SDEs, and SSAs. In this paper, we demonstrate that using these solvers, Catalyst's simulation speed often outperforms the other tools by more than one order of magnitude. Catalyst has the ability to include Julia-native functions within rate laws and stoichiometric expressions, and to include coupled ODE or algebraic constraint equations for reaction rate equation models (potentially resulting in *differential-algebraic equation* (DAE) models). For example, to encode bursty reactions stoichiometric coefficients can be defined using standard Julia functions that sample from a random variable distribution. Similarly, rate-laws can include data-driven modeling terms (e.g. neural networks) constructed via Julia libraries such as Surrogates.jl, SciMLSensitivity.jl, and DiffEqFlux.jl. Moreover, Catalyst generates differentiable models, which can be easily incorporated into higher-level Julia codes that require automatic differentiation [34] and composed with other Julia libraries. One current limitation of Catalyst is that in contrast to BioNetGen, COPASI, and GillesPy2, Catalyst can not generate inputs for hybrid and $\tau$-leaping solvers, though adding support for these features is planned.

In the next sections we overview a basic workflow for using Catalyst to define and simulate CRNs; overview how Catalyst performs relative to several popular CRN modeling packages for solving ODEs and simulating stochastic chemical kinetics models; discuss Catalyst's symbolic representation of CRNs, Catalyst's network analysis functionality, and how it can compose with other Julia packages; and introduce some of the higher-level applications in which Catalyst models can be easily embedded.

## 2 Results

### 2.1 The Catalyst DSL enables models to be created using chemical reaction notation

Catalyst offers several ways to define a CRN model, with the most effortless option being the `@reaction_network` DSL. This feature extends the natural Julia syntax via a macro, allowing users to declare CRN models using classic chemical reaction notation (as opposed to declaring models using equations, or by declaring reactions implicitly or through functions). This alternative notation makes scripts more human-legible, and greatly reduces code length (simplifying both script writing and debugging). Using the DSL, the CRN's chemical reactions are listed, each preceded by its reaction rate (Fig 1). From this, the system's species and parameters are automatically extracted and a `ReactionSystem` IR structure is created (which can be used as input to e.g. numerical simulators).

To facilitate a more concise notation, similar reactions (e.g. several degradation events) can be bundled together. Each reaction rate can either be a constant, a parameter, or a function. Predefined Michaelis–Menten and Hill functions are provided by Catalyst, but any user-defined Julia function can be used to define a rate. Both parametric and non-integer stoichiometric coefficients are possible. There are also several non-DSL methods for model creation. They include loading networks from files via SBMLToolkit.jl [36] (for SBML files) and ReactionNetworkImporters.jl [37] (for BioNetGen generated .net files). CRNs can also be created via defining symbolic variables via the combined Catalyst/ModelingToolkit API, and directly building `ReactionSystem`s from collections of `Reaction` structures. This enables programmatic definition of CRNs, making it possible to create large models by iterating through a relatively small number of rules within standard Julia scripts.

### 2.2 Catalyst models can be simulated using a wide range of high-performance methods

Numerical simulations of Catalyst models are generally carried out using the DifferentialEquations.jl package [31]. It contains a large number of numerical solvers and a wide range of additional features (such as event handling, support for GPUs and threading, flexibility in choice of linear solvers for stiff integrators, and more). The package is highly competitive, often

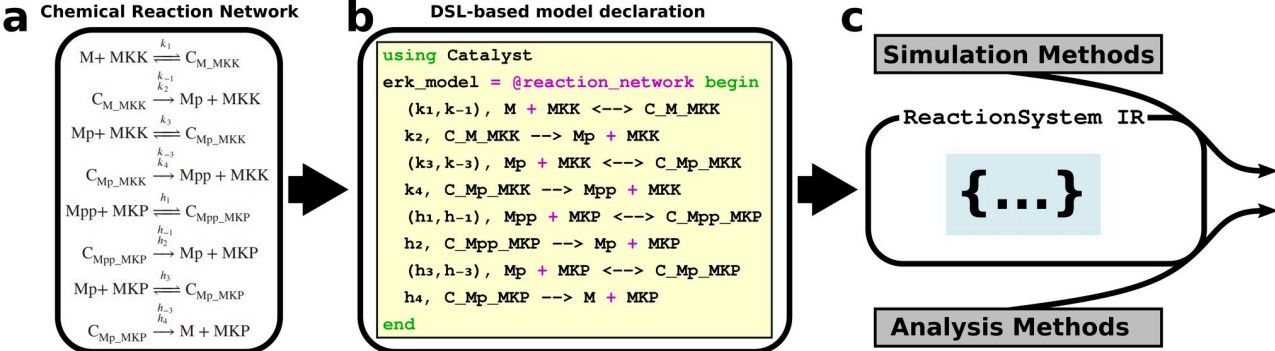

**Fig 1. Catalyst connects an intuitive domain-specific language with a well-supported intermediate representation.** The extracellular signal-regulated kinase (ERK) network is important to the regulation of many cellular functions, and its disruption has been implicated in cancer [35]. (a) A CRN representation of the ERK network. (b) A model of the ERK CRN can be implemented in Julia through the Catalyst DSL, using code very similar to the actual CRN representation. (c) From this code, the DSL generates a `ReactionSystem` intermediate representation (IR) that is the target structure for a range of supported simulation and analysis methods.

outperforming packages written in C and Fortran [31]. Simulation syntax is straightforward, and output solutions can be plotted using the Plots.jl package [38] via a recipe that allows users to easily select the species and times to display. CRNs can be translated and simulated using the ODE-based RREs, the SDE-based CLE, and through discrete SSAs (Fig 2).

To demonstrate the performance of these solvers, we benchmarked simulations of CRN models using a range of CRN modeling tools (BioNetGen, Catalyst, COPASI, GillesPy2, and Matlab's SimBiology toolbox). These tools were selected as they are popular and highly cited, well documented, scriptable for running benchmark studies, and actively maintained. The Matlab SimBiology toolbox was selected due to the enduring popularity of the Matlab language. Overall, they provide a representative sample of the broader chemical reaction network modeling software ecosystem. We used both ODE simulations and discrete SSAs. Fewer packages permit SDE simulations, hence such simulations were not benchmarked. We note, however, that DifferentialEquations' SDE solvers are highly performative [42]. When comparing a range of models, from small to large, we see that Catalyst typically outperforms the other packages, often by at least an order of magnitude (Fig 3). For the ODE benchmarks, to try to provide as fair a comparison as possible, identical absolute and relative tolerances were used for all simulations. Furthermore, in Fig C in S1 Text we demonstrate the relation between simulation time and actual error across the Julia solvers, lsoda, and CVODE (with the native Julia solvers typically having smaller errors as compared to lsoda and CVODE for any given tolerance). All SSA methods tested generate exact realizations, in the sense that they should each give statistics consistent with the underlying Chemical Master Equation of the model [43], and their simulation times are hence directly comparable. Here, the wide range of methods provided by the JumpProcesses.jl package [44], a component of DifferentialEquations, enables Catalyst to outperform the other packages (most of which only provide Gillespie's direct method or its sorting direct variant [45]).

In contrast to the exact SSA methods, timestep-based ODE integrators typically provide a variety of numerical parameters, such as error tolerances and configuration options for implicit solvers (i.e. how to calculate Jacobians, how to solve linear and nonlinear systems, etc). ODE simulation performance then depends on which combinations of options are used with a given solver. Here, we limit ourselves to trying combinations of numeric solvers (Julia-native solvers for comparing performance of Catalyst-generated models, and lsoda and/or CVODE for comparisons between tools), methods for Jacobian computation and representation (automatic differentiation, finite differences, or symbolic computation, and dense vs. sparse representations), linear solvers (LU, GMRES, or KLU), and whether to use a preconditioner or not when using GMRES. The non-Catalyst simulators generally provide limited ability to change these options, in which case only the default was used in benchmarking. In contrast, the DifferentialEquations.jl solvers that Catalyst utilise, while they do not require the user to set these options, do give them full control to do so. Full documentation is available at [46]. The details of the most performant options we used for each tool and model are provided in Section 4.1. A list of all benchmarks we carried out (for various combinations of tool, method, and options) is provided in Section B in S1 Text, with their results described in Figs A and B in S1 Text. Finally, the benchmarking process is described in more detail in Section 4.1.

The observed performance results for Catalyst-generated models arise from a variety of factors. For example, Catalyst inlines all mass action reaction terms for ODE models within a single generated function that evaluates the ODE derivative. This provides opportunities for the compiler to optimise expression evaluation, and avoids the overhead of repeatedly calling non-inlined functions to evaluate such terms. For the largest ODE models, Catalyst and Modeling-Toolkit's support for generating explicit sparse Jacobians led to significant performance improvements when using the CVODE solver, see Section 4.1 and S1 Text. For jump process

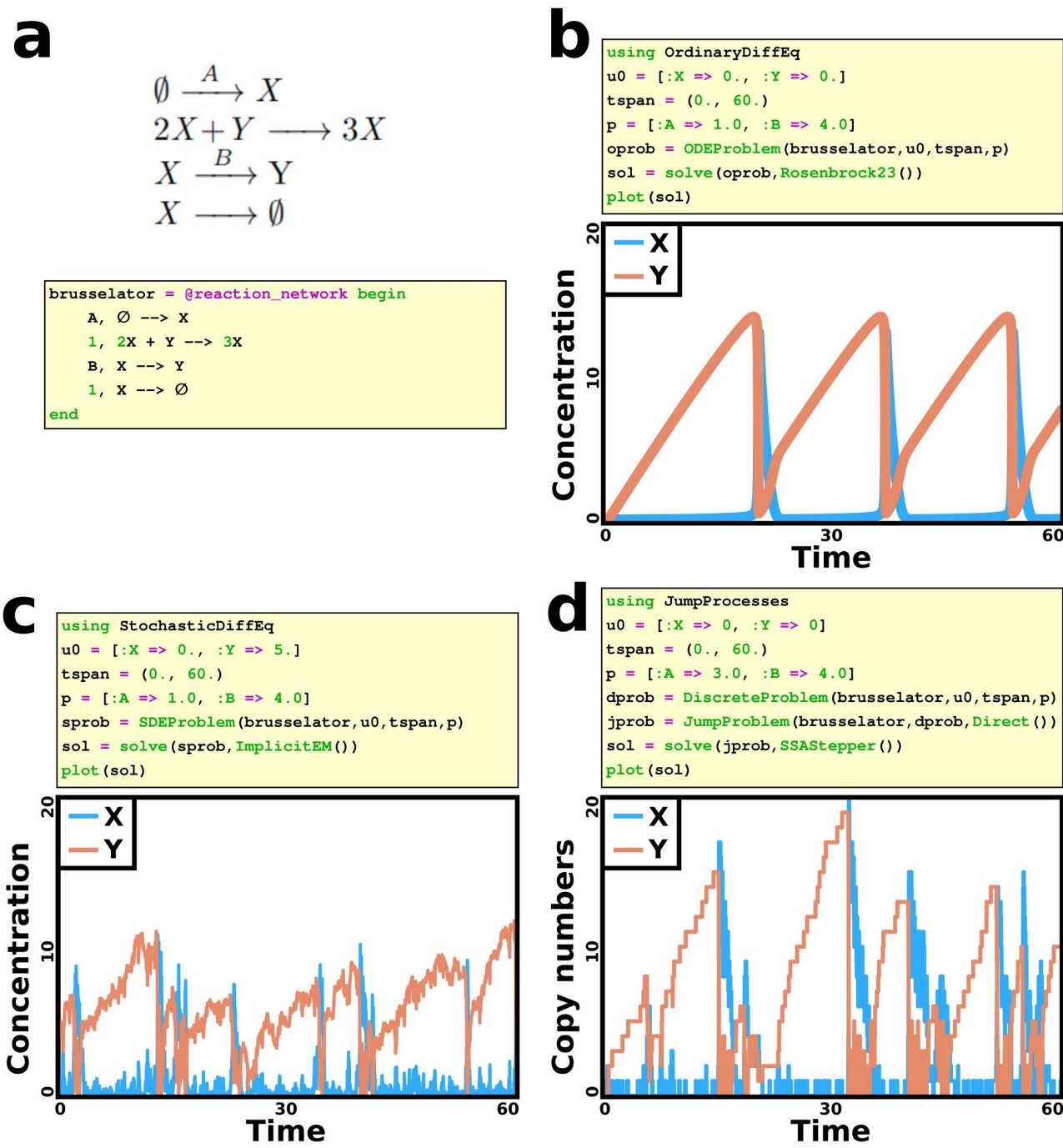

**Fig 2. Catalyst models can be simulated using both deterministic and stochastic interpretations.** (a) The Brusselator network contains two species (X and Y) and two parameters (A and B, in practical implementation these are species present in excess, but they can in practice be considered parameters) [39, 40]. Here, we show the four reactions of the Brusselator CRN, and its implementation using the Catalyst DSL. (b-d) Simulations of models for the Brusselator at the three physical scales supported by Catalyst (RRE, CLE, SSA). Post-processing has been carried out on the plots to improve their visualization in this article's format. (b) While $B > 1 + A^2$, the deterministic model exhibits a limit cycle. This is confirmed using ODE RRE simulations. (c) The model can also be simulated using the stochastic CLE interpretation. (d) Finally, the discrete, stochastic, jump process interpretation is simulated via Gillespie's direct method. The system displays a limit cycle even though $B < 1 + A^2$, confirming the well known phenomenon of noise induced oscillations [41].

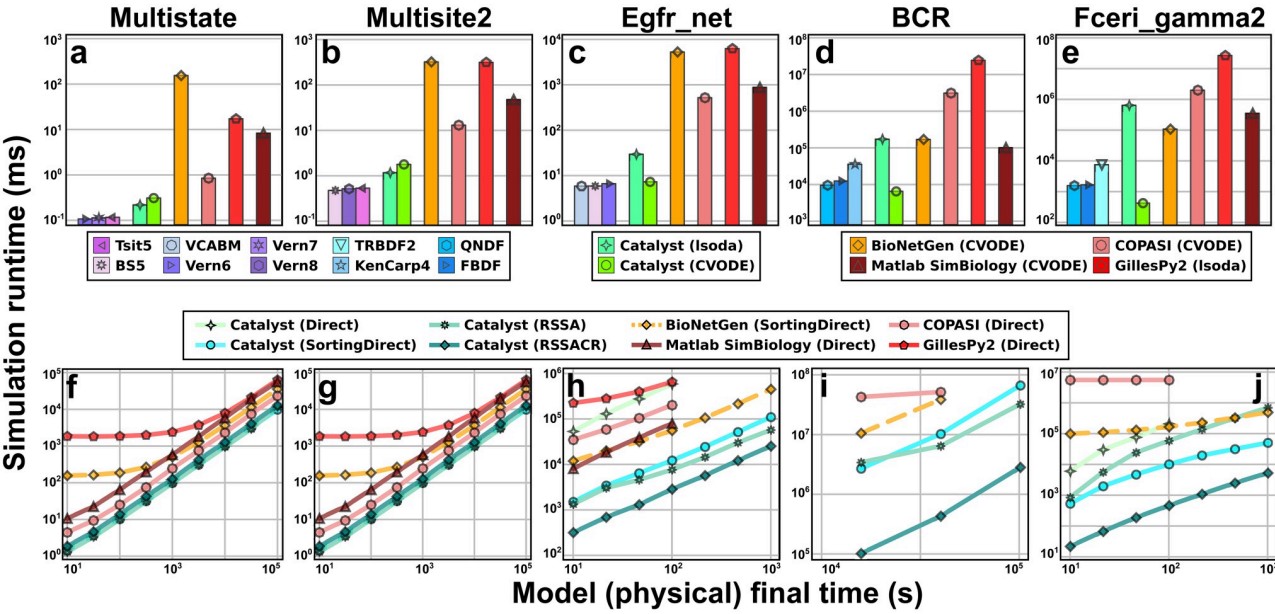

**Fig 3. Simulations of Catalyst models outperform those of other modeling packages.** Benchmarks of simulation runtimes for Catalyst and four other modeling packages (BioNetGen, COPASI, GillesPy2, and Matlab SimBiology). The benchmarks were run on the multi-state (Multistate, 9 species and 18 reactions [47]), multi-site (Multisite 2, 66 species and 288 reactions [48]), epidermal growth factor receptor signalling (Egfr_net, 356 species and 3749 reactions [49]), B-cell receptor (1122 species and 24388 reactions [50]), and high-affinity human IgE receptor signalling (Fceri_gamma2, 3744 species and 58276 reactions [51]) models. (a-e) Benchmarks of deterministic RRE ODE simulations of the five models. Each bar shows, for a given method, the runtime to simulate the model (to steady-state for those that approach a steady-state). For Catalyst, we show the three best-performing native Julia methods, as well as the performance of lsoda and CVODE. For each of the other tools, we show its best-performing method. Identical values for absolute and relative tolerance are used across all packages and methods. For each benchmark, the method options used can be found in Section 4.1, the exact benchmark times in Table A in S1 Text, and further details on the solver options for each tool in Section B in S1 Text. While this figure only contains the most performant methods, a full list of methods investigated can be found in Section B in S1 Text, with their results described in Figs A and B in S1 Text. (f-j) Benchmarks of stochastic chemical kinetics SSA simulations of the five models. Via JumpProcesses.jl, Catalyst can use several different algorithms (e.g. Direct, Sorting Direct, RSSA, and RSSACR above) for exact Gillespie simulations. Here, the simulation runtime is plotted against the (physical) final time of the simulation. Due to their long runtimes, some tools were not benchmarked for the largest models. We note that, in [52], it was remarked that BioNetGen (dashed orange lines) use a pseudo-random number generator in SSAs that, while fast, is of lower quality than many (slower) modern generators such as Mersenne Twister. For full details on benchmarks, see Section 4.1.

SSA simulations, Catalyst uses a sparse reaction specification that automatically analyses each reaction, and then classifies the reaction into the most performant but physically valid representation supported by JumpProcesses.jl (corresponding to jumps with general time-varying intensities, jumps with general rate expressions but for which the intensity is constant between the occurrence of two jumps, and jumps for which the intensity is a mass action type rate law). This enables JumpProcesses.jl to avoid the overhead of calling a large collection of user-provided functions via pointers, by using a single pre-defined and inlined function to evaluate individual mass action reaction intensities, while still supporting calling general user rate functions via pointers (for non-mass action rate laws). These Catalyst-specific features, when coupled to the large variety of solvers in DifferentialEquations.jl and broad flexibility in tuning solver components (i.e. different Jacobian and jump representations, flexibility in choice of linear solvers, etc.), help enable Catalyst's observed performance.

## 2.3 Catalyst enables composable, symbolic modeling of CRNs

Catalyst's primary feature is that its models are represented using a CAS, enabling them to be algebraically manipulated. Examples of how this is utilised include automatic computation

of system Jacobians, calculation and elimination of conservation laws, and simplification of generated symbolic DAE models via ModelingToolkit's symbolic analysis tooling. These techniques can help speed up numeric simulations, while also facilitating higher level analysis. One example is enabling users to generate ODE models with non-singular Jacobians via the elimination of conservation laws, which can aid steady-state analysis tooling. Catalyst also provides a network analysis API, enabling the calculation of a variety of network properties beyond conservation laws, including linkage classes, weak reversibility, and deficiency indices.

Catalyst's symbolic representation permits model internals to be freely extracted, investigated, and manipulated, giving the user full control over their models (Fig 4). This enables various forms of programmatic model creation, extension and composition. Model structures that occur repetitively can be duplicated, and disjoint models can be connected together. For example, such functionality can be used to model a population of cells, each with defined neighbours, where each cell can be assigned a duplicate of the same simple CRN. The CRNs within each cell can then be connected to those of its neighbours, enabling models with spatial structures. Similarly, one could define a collection of genetic modules, and then compose such modules together into a larger gene regulatory network.

Catalyst is highly flexible in the allowed Julia functions that can be used in defining rates, rate laws, or stoichiometry coefficients. This means that while reaction rates and rate laws are typically constants, parameters, or simple functions, e.g. Hill functions, they may also include other terms, such as neural networks or data-driven, empirically defined, Julia functions. Likewise, stoichiometric coefficients can be random variables by defining them as a symbolic variable, and setting that variable equal to a Julia function sampling the appropriate probability distribution. Such functionality can be utilized, for example, to model transcriptional bursting [53], where the produced mRNA copy-numbers are random variables. Finally, standard Catalyst-generated ODE and SDE models are differentiable, in that the generated codes can be used in higher-level packages that rely on automatic differentiation [34]. In this way Catalyst-generated models can be used in machine-learning based analyses.

That Catalyst gives full access to its model internals, combined with its composability, allows other packages to easily integrate into, and build upon, it. Indeed, this is already being utilised by independent package developers. The MomentClosure.jl Julia package, which implements several techniques for moment closure approximations, is built to be deployed on Catalyst models [54]. It can generate symbolic finite-dimensional ODE system approximations to the full, infinite system of moment equations associated with the chemical master equation. These symbolic approximations can then be compiled and solved via ModelingToolkit in a similar manner to how Catalyst's generated RRE ODE models are handled. Similarly, FiniteStateProjection.jl [55] builds upon Catalyst and ModelingToolkit to enable the numerical solution of the chemical master equation, while DelaySSAToolkit.jl [56] can accept Catalyst models as input to its SSAs that handle stochastic chemical kinetics models with delays. Another example of how Catalyst's flexibility enables its integration into the Julia ecosystem is that CRNs with polynomial ODEs (a condition that holds for pure mass action systems) can be easily converted to symbolic steady-state systems of polynomial equations. This enables polynomial methods, such as homotopy continuation, to be employed on Catalyst models. Here, homotopy continuation (implemented by the HomotopyContinuation.jl Julia package) can be used to reliably compute all roots of a polynomial system [57]. This is an effective approach for finding multiple steady states of a system. When the CRN contains Hill functions (with integer exponents), by multiplying by the denominators, one generates a polynomial system with identical roots to the original, on which homotopy continuation can still be used.

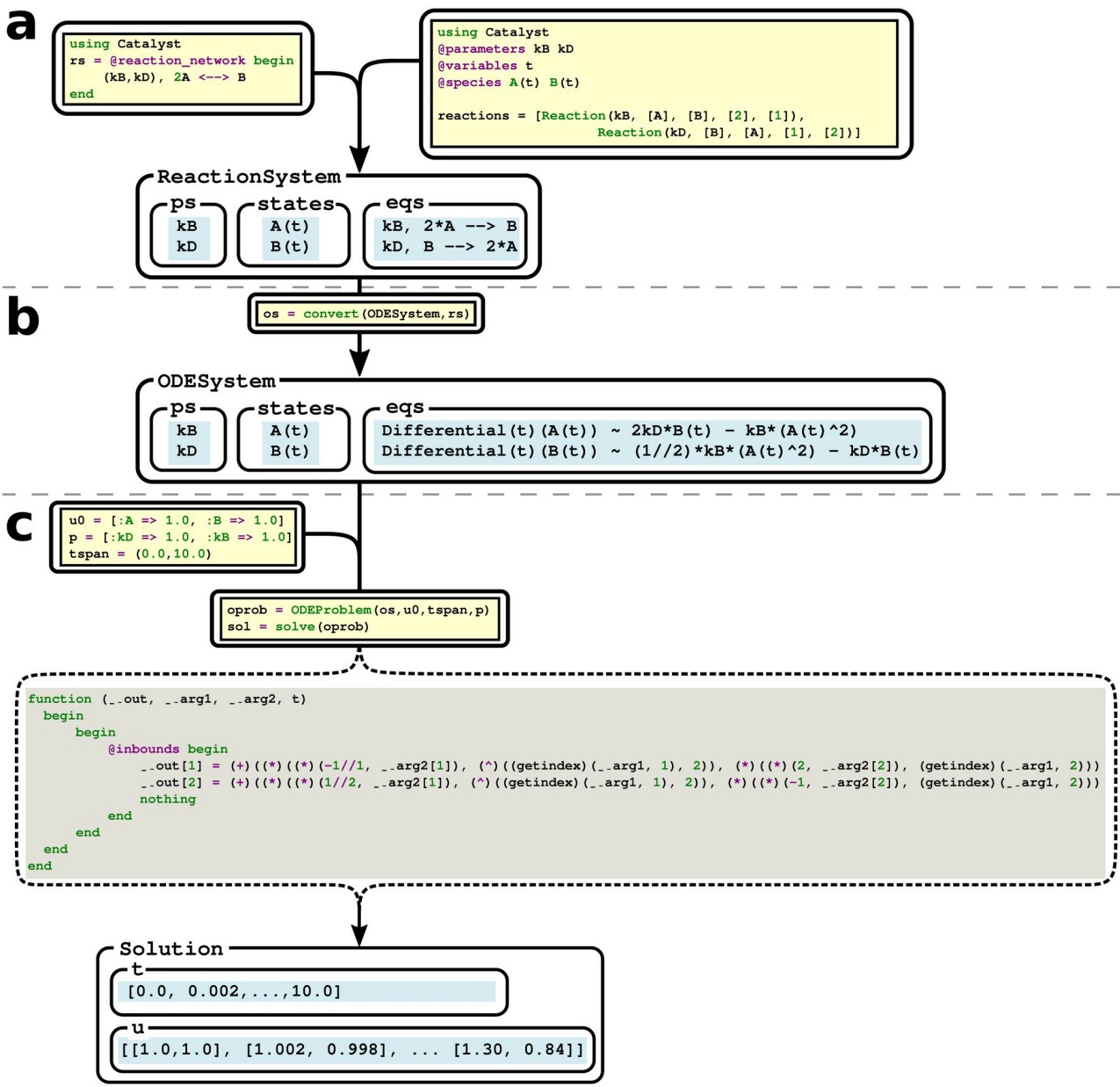

**Fig 4. The simulation pipeline of a Catalyst model, with internal intermediates displayed.** Code as written by the user (yellow background), and as generated internally by Catalyst and ModelingToolkit (blue and grey backgrounds respectively) are shown, in addition to the generated structures and their fields (blue background, some of the internal fields are omitted in all displayed structures). (a) A symbolic `ReactionSystem` for a reversible dimerisation reaction is created using either the DSL, or programmatically using the Symbolics computer algebra system. (b) The `ReactionSystem` can be converted into a ModelingToolkit `ODESystem` structure, corresponding to a symbolic RRE ODE model. (c) By providing initial conditions, parameter values, and a time span, the `ODESystem` can be simulated, generating an output solution. The generated (internal) Julia code for evaluating the derivatives defining the ODEs, which gets compiled and is input to the ODE solver, is displayed in grey. At each step, the user has the ability to investigate and manipulate the generated structures.

## 2.4 Catalyst models are compatible with a wide range of ancillary tools and methods

The Julia SciML, and broader Julia, ecosystem offers a wide range of techniques for working with models and data based around the IR that Catalyst produces (Fig 5). While the reactions

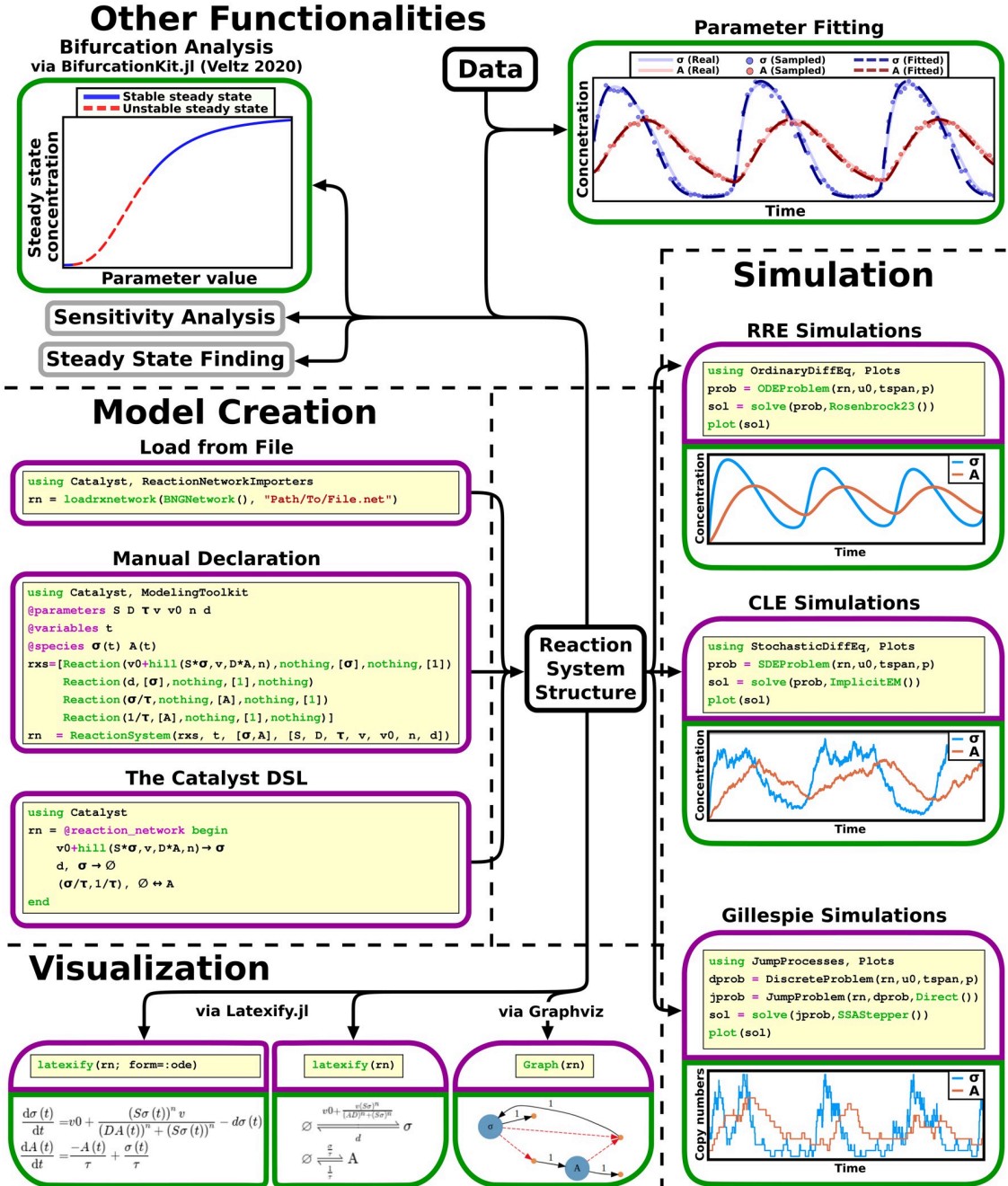

**Fig 5. A wide range of features are available for Catalyst model analysis.** A CRN model can be created either through the DSL, by manually declaring the reaction events, or by loading it from a file. The model is stored in the `ReactionSystem` IR, which can be used as input to a wide range of methods. Purple boxes indicate code written by the user, and green boxes the corresponding output. For some methods, either one, or both, boxes are omitted.

that constitute a CRN are often known in developing a model, system parameters (these typically correspond to reaction rates) rarely are. A first step in analyzing a model is identifiability analysis, where we determine whether the parameters can be uniquely identified from the data [58]. This is enabled through the StructuralIdentifiability.jl package. In the next step,

parameters can be fitted to data. This can be done using DiffEqParamEstim.jl, which provides simple functions that are easy to use. Alternatively, more powerful packages, like Optimization.jl and the Turing.jl Julia library for Bayesian analysis, offer increased flexibility for experienced users [59]. Furthermore, unknown CRN structures (such as a species's production rate) can be approximated using neural networks and then fitted to data. This functionality is enabled by the SciMLSensitivity package [60]. More broadly, system steady states can be computed using the NLSolve.jl or HomotopyContinuation.jl Julia packages [57]; bifurcation structures can be calculated, and bifurcation diagrams generated, with the BifurcationKit.jl library [61]; and SciMLSensitivity.jl and GlobalSensitivity.jl can be used to investigate the sensitivity and uncertainty of model solutions with regard to parameters [62]. Finally, options for displaying CRNs, either as network graphs (via Graphviz) or Latex formatted equations (via Latexify.jl), also exist.

## 3 Discussion

In this article, we have introduced the Catalyst library for modeling of CRNs. It represents models through the ModelingToolkit.jl IR, which is used across the SciML organization and Julia ecosystem libraries, and can be automatically translated into optimized inputs for numerical simulations (RRE ODE, CLE SDE, and stochastic chemical kinetics jump process models). Our benchmarks demonstrate that Catalyst often outperforms other tools by an order of magnitude or more. Moreover, it can compose with a variety of other Julia packages, including data-driven modeling tooling (parameter fitting and model inference), and other functionality (identifiable analysis, sensitivity analysis, steady state analysis, etc). The IR is based on the Symbolics.jl CAS, enabling algebraic manipulation and simplification of Catalyst models. This can both be harnessed by the user (e.g. to create models programmatically) and by software (e.g. for automated Jacobian computations). Finally, this also enables easy connection to other Julia packages for symbolic analysis, such as enabling polynomial methods (e.g homotopy continuations) to act on CRN ODEs that have a polynomial form.

In addition to the wide range of powerful tools enabled by the combination of the ModelingToolkit IR and the Symbolics CAS, Catalyst also provides a DSL that simplifies the declaration of smaller models. Of a finalized pipeline that evaluates a model with respect to a specific scientific problem, the model declaration is typically only a minor part. However, reaching a final model often requires the production and analysis of several alternative network topologies. If the barrier to create, or modify, a model can be reduced, more topologies can be explored in a shorter time. Thus, an intuitive interface can greatly simplify the model exploration portion of a research project. By providing a DSL that reads CRN models in their most natural form, Catalyst helps to facilitate model construction. In addition, this form of declaration makes code easier to debug, as well as making it easier to understand for non-experts.

While several previous tools for CRN modeling have been primarily designed around their own interface, we have instead designed Catalyst to be called from within standard Julia programs and scripts. This is advantageous, since it allows the flexibility of analysing a model with custom code, without having to save and load simulation results to and from files. Furthermore, by integrating our tool into a larger context (SciML), support for a large number of higher-order features is provided, without requiring any separate implementation within Catalyst. This strategy, with modeling software targeting an IR (here provided by ModelingToolkit) enables modelers across widely different domains to collaborate in the development and maintenance of tools. We believe this is the ideal setting for a package like Catalyst.

Development of Catalyst is still active, with several types of additional functionality planned. This includes specialised support for spatial models, including spatial SSA solvers for

the reaction-diffusion master equation, and general support for reaction models with transport on graphs at both the ODE and jump process level. A longer-term goal is to enable the specification of continuous-space reaction models with transport, and interface with Julia partial differential equation libraries to seamlessly generate such spatially-discrete ODE and jump process models. Furthermore, unlike BioNetGen, COPASI, and GillesPy2, Catalyst does not currently support hybrid methods. These allow model components to be defined at different physical scales (such as resolving some reactions via ODEs and others via jump processes) [63, 64]. This, as well as $\tau$-leaping-based solvers [65, 66], are planned for future updates. Such hybrid approaches can help to overcome the potential negativity of solutions that can arise in $\tau$-leaping and CLE-based models [67]. In the CLE case, Catalyst currently wraps rate laws within the coefficients of noise terms in absolute values to avoid square roots of negative numbers, allowing SDE solvers to continue time-stepping even when solutions become negative (following the approach in [68]). We hope to also integrate alternative modelling approaches, such as the constrained CLE [67], which avoid negativity of solutions via modification of the dynamics at the positive-negative population boundary. Finally, given Catalyst's support for units we hope to implement functionality for automatically converting between concentration and "number of" units within system specifications by allowing users to specify compartments with associated size units.

Catalyst is available for free under the permissive MIT License. The source code can be found at https://github.com/SciML/Catalyst.jl. It is also a registered package within the Julia ecosystem and can be installed from within a Julia environment using the commands `using Pkg; Pkg.add("Catalyst")`. Full documentation, including tutorials and an API, can be found at https://catalyst.sciml.ai/stable/. Issues and help requests can be raised either at the Catalyst GitHub page, on the Julia discourse forum (https://discourse.julialang.org/), or at the SciML organization's Julia language Slack channels (`#diffeq-bridged` and `#sciml-bridged`). The library is open to pull requests from anyone who wishes to contribute to its development. Users are encouraged to engage in the project.

# 4 Materials and methods

## 4.1 Benchmarks

Benchmarks were carried out using the five CRN models used in [52]. The .bngl files provided in [52] were used as input to BioNetGen, while COPASI, GillesPy2, and Matlab used the corresponding (BioNetGen generated) .xml files. Catalyst used the corresponding (by BioNetGen generated) .net files. The exception was the BCR model, for which we used the .bngl file from [50], rather than the one from [52]. Throughout the simulations, no observable values were saved. Where options were available to reduce solution time point save frequency, and these improved performances, these were used (Section C in S1 Text). BioNetGen, COPASI, and GillesPy2 simulations were performed using their corresponding Python interfaces. To ensure the correctness of the solvers, for each combination of model, tool, method, and options, ODE and SSA simulations were carried out and the results were plotted. The plots were inspected to ensure consistency across all simulations (Figs D-M in S1 Text). Runtimes were measured using `timeit` (in Python), BenchmarkTools.jl (in Julia, [69]), and `timeit` (in Matlab). For each benchmark, the median runtime over several simulations was used (the number of simulations carried out for each benchmark, over which we took the median, is described in Table 1).

For ODE benchmarks, simulation run times were measured from the initial conditions used in [52] to the time for the model to reach its (approximate) steady state (Table 2). The exception was the BCR model, which exhibited a pulsing limit cycle behaviour. For this, we

**Table 1. Number of simulations used to calculate median simulation times.**

| Model: | Multistate | Multisite2 | Egfr_net | BCR | Fceri_gamma2 |
|---|---|---|---|---|---|
| BioNetGen (ODE) | 10 | 10 | 10 | 10 | 10 |
| BioNetGen (SSA) | 10 | 10 | 10 | 4 | 5 |
| COPASI (ODE) | 10 | 10 | 10 | 10 | 10 |
| COPASI (SSA) | 10 | 10 | 10 | 2 | 5 |
| GillesPy2 (ODE) | 10 | 10 | 10 | 5 | 10 |
| GillesPy2 (SSA) | 10 | 10 | 10 | 2 | 5 |

For each benchmark, we performed a number of simulations, computing their median runtime. The number of such simulations depends on the tool and model (with this number given in this table). As default, we used 10, but in some cases we needed to reduce this to enable the benchmark to be completed within a reasonable time. For Julia and Matlab benchmarks, the number of simulations was automatically determined by the `timeit` tool and the BenchmarkTools.jl package, respectively.

simulated it over 20,000 time units, allowing it to complete three pulse events (Fig G in S1 Text). For ODE simulations, for all tools, the absolute tolerance was set to $10^{-9}$ and the relative tolerance $10^{-6}$. Primarily tests were carried out using the lsoda and CVODE solvers [70, 71]. However, Catalyst has access to additional ODE solvers via DifferentialEquations.jl (more specifically OrdinaryDiffEq.jl). Some of these (such as QNDF and TRBDF2) are competitive with lsoda and CVODE, hence these additional solvers were also benchmarked [72, 73]. All benchmarks were carried out on the MIT supercloud HPC [74]. We used its Intel Xeon Platinum 8260 units (each node has access to 192 GB RAM and contains 48 cores, of which only a single one was used). Each benchmark was carried out on a single, exclusive, node, to ensure they were not affected by the presence of other jobs. Julia, Matlab, and Python all were set to use only a single thread, ensuring multi-threading did not affect performance (e.g. Julia solvers will automatically utilise additional available threads to speed up the linear solvers of implicit simulators). Finally, work-precision diagrams were investigated to determine the relationship between simulation time and error in the native Julia solvers (Fig C in S1 Text). All benchmarking code is avaiable at (Code availability) under a permissive MIT license.

When using CVODE or implicit solvers, Catalyst permits a range of simulation options. By default, Jacobians are computed through automatic differentiation [34]. This option can either be disabled (with the Jacobian then being automatically computed through finite differences), or an option can be set to automatically compute, and use, a symbolic Jacobian from Catalyst models. Another option enables a sparse representation of the Jacobian matrix. Furthermore, the underlying linear solver for all implicit methods can be specified. We tried both the default option (which automatically selects one), but also specified either the LapackDense (using LU), GMRES, or KLU linear solvers. When the GMRES linear solver is used, a preconditioner can be set. Here we investigated both using no preconditioner, and using an incomplete LU preconditioner (described further in Section B in S1 Text).

**Table 2. Final (physical) time for model steady states in ODE benchmarks.**

| Model: | Multistate | Multisite2 | Egfr_net | BCR | Fceri_gamma2 |
|---|---|---|---|---|---|
| | 20 s | 2 s | 10 s | 20,000 s | 150 s |

For each model, we determined the time at which it had (approximately) reached a steady state. These times were used for the ODE benchmarks in Fig 3 and Fig A in S1 Text. Unlike the other models, BCR exhibits a limit cycle. Here, rather than simulating until an (approximate) steady state had been reached, we simulated it for 20,000 time units (permitting it to complete 3 pulse events, Fig G in S1 Text).

Jacobians were generated using either automatic differentiation (when either the Multistate, Multisite2, or Egfr_net models were simulated using Julia solvers) or finite differences. The exception was for the KLU linear solver, for which we used a symbolically computed Jacobian. When we used either the KLU linear solver, or preconditioned GMRES, a sparse Jacobian representation was used. Generally, the non-Catalyst tools have fewer available solvers (typically depending on CVODE) and options, however, we tried those we found available. We also note that Catalyst CVODE simulations without any options specified still compare favourably to the other tools (Fig A in S1 Text). The methods and options used for the benchmarks in Fig 3 are described in Table 3. Their performance is also described in Table A in S1 Text (this contains the same information as Fig 3, but as numbers rather than a bar chart). For a full list of benchmarks carried out, and the options used, see Section B in S1 Text. Furthermore, Fig A in S1 Text shows the performance of all trialed combinations of methods and options, with Fig B in S1 Text showing the performance when the simulations are carried out for increasing final model (physical) times.

Stochastic chemical kinetics simulations of Catalyst models used SSAs defined in JumpProcesses.jl [44], a component of DifferentialEquations.jl. In Fig 3, Direct refers to Gillespie's direct method [9], SortingDirect to the sorting direct method of [45], RSSA and RSSACR to the rejection and composition-rejection SSA methods of [75–77]. Dependency graphs needed for the different methods are automatically generated via Catalyst and ModelingToolkit as input to the JumpProcesses.jl solvers. Due to supercloud not permitting single runs longer than 4 days, for the largest models, the slowest tools and methods were not benchmarked. The BCR model exhibits pulses, to ensure that at least some pulses were included in each SSA simulation, this model was simulated over very long timespans ($> 10,000$ seconds). For a full list of SSA benchmarks and their options, please see Section C in S1 Text.

The benchmarks were carried out on Julia version 1.8.5, using Catalyst version 13.1.0, JumpProcesses version 9.5.1, and OrdinaryDiffEq version 6.49.0. Note that JumpProcesses and OrdinaryDiffEq are both components in the meta DifferentialEquations.jl package. We used Python version 3.9.15, the version 0.7.9 python interface for BioNetGen, the basico

**Table 3. Options used for the benchmarked ODE methods displayed in the main text figure.**

| Model: | Multistate | Multisite2 | Egfr_net | BCR | Fceri_gamma2 |
|---|---|---|---|---|---|
| Julia solver 1 | Vern6 | BS5 | VCABM | QNDF[1,2,3,4] | QNDF[1,2,3,4] |
| Julia solver 2 | Vern7 | Vern8 | BS5 | FBDF[1,2,3,4] | FBDF[1,2,3,4] |
| Julia solver 3 | Tsit5 | Tsit5 | Vern6 | KenCarp4[1,2,3,4] | TRBDF2[1,2,3,4] |
| Catalyst lsoda | lsoda | lsoda | lsoda | lsoda | lsoda |
| Catalyst CVODE | CVODE | CVODE[1] | CVODE[1] | CVODE[1,2,4] | CVODE[1,2,4] |
| BioNetGen | CVODE[1] | CVODE[1] | CVODE[1] | CVODE | CVODE[1] |
| COPASI | CVODE | CVODE | CVODE | CVODE | CVODE |
| GillesPy2 | lsoda | lsoda | lsoda | lsoda | lsoda |
| Matlab | CVODE | CVODE | CVODE | CVODE | CVODE |

For each model the options used for the 3 most performant native Julia solvers, the Julia lsoda and CVODE implementations, and each other tool (the results using these benchmarks are found in Fig 3). Each field contains the method used for that model. Further options (including whenever a specific linear solver was selected) are described through superscript tags.

[1] GMRES linear solver was used.

[2] Sparse Jacobian representation was used (a Catalyst option only).

[3] Automatic differentiation (as a mean of Jacobian calculation) was turned off (a Catalyst option only).

[4] An incomplete LU preconditioner was supplied to the GMRES linear solver (a Catalyst option only).

version 4.47 python interface for COPASI, GillesPy2 version 1.8.1, and Matlab version 9.8 with SimBiology version 5.10.

## Supporting information

**S1 Text. Additional benchmarks and benchmark information.**
(PDF)

## Acknowledgments

The authors thank the 26 other individuals who contributed commits to Catalyst, the Catalyst tutorials, and the Catalyst documentation, along with the many users who have offered suggestions and opened issues. The authors acknowledge the MIT SuperCloud and Lincoln Laboratory Supercomputing Center for providing (HPC, database, consultation) resources that have contributed to the research results reported within this paper/report.

## Author Contributions

**Conceptualization:** Torkel E. Loman, Chris Rackauckas, Samuel A. Isaacson.

**Formal analysis:** Torkel E. Loman, Chris Rackauckas, Samuel A. Isaacson.

**Funding acquisition:** Chris Rackauckas, Samuel A. Isaacson.

**Investigation:** Torkel E. Loman, Chris Rackauckas, Samuel A. Isaacson.

**Methodology:** Torkel E. Loman, Chris Rackauckas, Samuel A. Isaacson.

**Project administration:** Chris Rackauckas, Samuel A. Isaacson.

**Software:** Torkel E. Loman, Yingbo Ma, Vasily Ilin, Shashi Gowda, Niklas Korsbo, Nikhil Yewale, Chris Rackauckas, Samuel A. Isaacson.

**Supervision:** Chris Rackauckas, Samuel A. Isaacson.

**Validation:** Torkel E. Loman, Chris Rackauckas, Samuel A. Isaacson.

**Visualization:** Torkel E. Loman.

**Writing – original draft:** Torkel E. Loman, Chris Rackauckas, Samuel A. Isaacson.

**Writing – review & editing:** Torkel E. Loman, Chris Rackauckas, Samuel A. Isaacson.

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
