## [Decision Letter · Decision Letter 0]

15 Aug 2023

Dear Dr Loman,

Thank you very much for submitting your manuscript "Catalyst: Fast and flexible modeling of reaction networks" for consideration at PLOS Computational Biology. As with all papers reviewed by the journal, your manuscript was reviewed by members of the editorial board and by several independent reviewers. The reviewers appreciated the attention to an important topic. Based on the reviews, we are likely to accept this manuscript for publication, providing that you modify the manuscript according to the review recommendations.

Sincerely,

Christos A. Ouzounis

Academic Editor

PLOS Computational Biology

Kiran Patil

Section Editor

PLOS Computational Biology

Reviewer's Responses to Questions

**Comments to the Authors:**

Reviewer #1: In the manuscript, Loman et al. present Catalyst.jl, a Julia package for modelling and simulation of chemical reaction networks. They describe how Catalyst is integrated into the SciML package ecosystem, leveraging other packages for symbolic model representation and numerical simulations. The authors demonstrate how chemical reaction networks in Catalyst can be generated, simulated, and extended or analysed further using intrinsic Catalyst features, other SciML tools and third-party Julia packages. They also perform an extensive set of model simulation benchmarks using a variety of ODE and SSA solvers, comparing Catalyst’s runtime performance to that of several other popular modelling packages.

In my opinion Catalyst.jl is a great library that works beautifully in tandem with the broader Julia ecosystem for numerical simulation and higher-level analysis, making Julia the programming language of choice for many computational biologists. On GitHub, the package is well documented and includes thorough tutorials. The manuscript itself is clear to follow and does a good job in presenting Catalyst, showcasing its features and higher-level applications. I also appreciate the rigorous simulation benchmarks and their detailed descriptions.

In summary, Catalyst.jl is a great contribution to the field and I happily endorse the manuscript’s publication.

Nevertheless, I had a few minor comments/suggestions for the authors to potentially consider:

• “Data and Code Availability” in the additional info has an old URL that does not work, but the correct one is provided in the Code Availability section of the main text.

• There appears to be a reference missing in “That a CRN can be unambiguously represented using these models forms the basis of several CRN modeling tools [?,11–22]”. In addition, another software tool that perhaps should be cited given its list of features is ”CERENA: ChEmical REaction Network Analyzer—A Toolbox for the Simulation and Analysis of Stochastic Chemical Kinetics”, PLoS ONE 11(1): e0146732.

• In Figure 2, it might be helpful to show the Brusselator reaction diagram and assumptions made to make life easier for an unfamiliar reader, or at least mention it in the caption. Alternatively, a more accessible reference could be useful: from a quick look at the one cited, i.e., Lefever et al. (1988), it does not seem to clearly define the Brusselator as used here and does not explicitly discuss the limit cycle condition (relationship between A and B). Perhaps "Elements of Applied Bifurcation Theory" by Y. A. Kuznetsov or another textbook would work better.

• In Section 2.2, is there a particular reason why Catalyst is benchmarked against BioNetGen, COPASI, GillesPy2, Matlab’s SimBiology, and not any other packages? These seem to be chosen very reasonably given their popularity, list of features and them being actively maintained, but it might be worthwhile to mention this explicitly.

• Regarding Section 2.2 and Fig. 4, it might be interesting to have even a very short discussion summarising the potential main factors leading to such notable runtime differences between CVODE and lsoda in different languages/packages, as well as almost always superior performance of native Julia solvers. Is it mostly about native language performance, specific implementations of the solvers or other algorithmic choices made?

• In Section 2.4, it might be more fitting with the paragraph style to specify which Julia packages allow approximating the unknown CRN structures using neural networks.

• Would be interesting to hear more details about the planned updates for spatial model support mentioned in the Discussion. Would that be a compartmentalised approach (akin to reaction–diffusion master equation) and what specific spatial SSA solvers would it include? Or does this also imply support for continuous reaction-diffusion processes?

• One quality of life improvement in Catalyst would be allowing to associate a volume parameter with each compartment and hence automatically scale the mass-action reaction rates according to volume. On a different note, an interactive GUI even in a very limited form (tutorials/specific Pluto notebooks?) could make the software more appealing for biologists with little to no programming experience. I see that both these comments are to some extent covered in the issues on GitHub, but perhaps expanding on future work and touching upon these and other similar possible improvements would be an insightful addition.

Reviewer #2: In the manuscript entitled ‘Catalyst: Fast and flexible modeling of reaction networks’ by Loman et al the authors developed a tool, Catalyst.jl, to describe biochemical reaction networks in the framework of Julia programming language. The Julia library Catalyst.jl is a symbolic modelling package where users can create the network model using Catalyst's domain specific language (DSL). The models created using this framework can now be simulated to generate deterministic or stochastic trajectories using various types of methods available in the existing package in Julia (DifferentialEquations.jl). Furthermore models developed in the Catalyst can also be used for variety of other purposes (e.g. bifurcation analysis, parameter estimation) using existing tools under Julia library.

There are many types of standalone modelling tools (e.g. COPASI, BioNetGen etc.) available to systems biology researchers for simulating biochemical reaction networks. The main appealing factor of Catalyst, in my opinion, is that in can be integrated into diverse types of existing Julia programming tools to achieve the desired objective. I recommend publication of the manuscript in the Plos Computational Biology upon justification of the following points.

1. Often system biology models are phenomenological in nature where the reaction rates are nonlinear with phenomenological rate functions. Although the authors mentioned about Hill function, however it is not clear whether user can customize the rate function as needed.

2. Does Catalyst allow non-integer Hill coefficient?

3. Due to the gaussian nature of the noise, solution of chemical Langevin equation may lead to negative concentration/population if the copy number of the relevant species becomes very low. How this scenario is addressed in the Catalyst?

4. Is the computational efficiency due to the Catalyst or the differential equation solvers developed in the Julia language?

**Have the authors made all data and (if applicable) computational code underlying the findings in their manuscript fully available?**

Reviewer #1: Yes

Reviewer #2: Yes

PLOS authors have the option to publish the peer review history of their article (what does this mean?). If published, this will include your full peer review and any attached files.

Reviewer #1: No

Reviewer #2: No

Figure Files:

Data Requirements:

Reproducibility:

References:

---

## [Decision Letter · Decision Letter 1]

19 Sep 2023

Dear Dr Loman,

We are pleased to inform you that your manuscript 'Catalyst: Fast and flexible modeling of reaction networks' has been provisionally accepted for publication in PLOS Computational Biology.

Best regards,

Christos A. Ouzounis

Academic Editor

PLOS Computational Biology

Kiran Patil

Section Editor

PLOS Computational Biology

Reviewer's Responses to Questions

**Comments to the Authors:**

Reviewer #1: I thank the authors for the detailed responses to my questions and the changes made in response.

Reviewer #2: In the revised version of the manuscript the authors have addressed quires with satisfaction and thus I recommend its publication in the PLOS Computational Biology.

**Have the authors made all data and (if applicable) computational code underlying the findings in their manuscript fully available?**

Reviewer #1: Yes

Reviewer #2: Yes

PLOS authors have the option to publish the peer review history of their article (what does this mean?). If published, this will include your full peer review and any attached files.

Reviewer #1: No

Reviewer #2: No

---

## [Editor Report · Acceptance letter]

4 Oct 2023

PCOMPBIOL-D-23-00868R1 

Catalyst: Fast and flexible modeling of reaction networks

Dear Dr Loman,

I am pleased to inform you that your manuscript has been formally accepted for publication in PLOS Computational Biology. Your manuscript is now with our production department and you will be notified of the publication date in due course.

With kind regards,

Anita Estes
